# Clinical Effectiveness of Immune Checkpoint Inhibitors in Non-Small-Cell Lung Cancer with a Poor Performance Status

**DOI:** 10.3390/medicina57111273

**Published:** 2021-11-19

**Authors:** Kyoichi Kaira, Hisao Imai, Atsuto Mouri, Ou Yamaguchi, Hiroshi Kagamu

**Affiliations:** Department of Respiratory Medicine, Comprehensive Cancer Center, International Medical Center, Saitama Medical University, Saitama 350-1298, Japan; m06701014@gunma-u.ac.jp (H.I.); mouria@saitama-med.ac.jp (A.M.); ouyamaguchi@gmail.com (O.Y.); kagamu19@saitama-med.ac.jp (H.K.)

**Keywords:** ^18^F-FDG PET, PD-1 blockade, immunotherapy, lung cancer, long-term survival, metabolic response, visual assessment

## Abstract

Immune checkpoint inhibitors (ICIs) are standard treatments for patients with lung cancer. PD-1/PD-L1 or CTLA4 antibodies are chosen as the first-line therapy, contributing to the long-term survival and tolerability. Unlike molecular targeting agents, such as gefitinib, lung cancer patients with a poor performance status (PS) display unsatisfactory clinical improvements after ICI treatment. Several previous reports also demonstrated that the PS is identified as one of the most probable prognostic factors for predicting poor outcomes after ICI treatment. However, first-line pembrolizumab seemed to be effective for lung cancer patients with a PS of 2 if PD-L1 expression was greater than 50%. Currently, the induction of ICIs in patients with lung cancer with a poor PS is controversial. These problems are discussed in this review.

## 1. Introduction

Immune checkpoint inhibitors (ICIs), such as programmed death-1 (PD-1)/programmed death ligand-1 (PD-L1) and cytotoxic T-lymphocyte-associated protein 4 (CTLA4) antibodies, are widely administered to patients with several types of cancers. In particular, it is surprising that long-term survival of more than 5 years was observed in patients with advanced malignant melanoma and metastatic/recurrent non-small-cell lung cancer (NSCLC) after the initiation of PD-1 blockade monotherapy [1,2]. Thus, we believe that a PD-1 blockade may also bring clinical benefits to cancer patients with a poor performance status (PS). Unfortunately, several previous reports demonstrated that PD-1 blockade monotherapy was not effective in such patients [3,4,5]. As a first-line setting, chemotherapeutic regimens, including a PD-1 blockade, are universally established as standard treatment for patients with advanced NSCLC without any driver mutations. Unlike molecular targeting agents, such as gefitinib, advanced or recurrent NSCLC patients with poor PSs displayed unsatisfactory clinical improvements after ICI treatment. Several previous reports also showed that the PS is one of the most important prognostic factors for predicting poor outcomes after ICI treatment [3,4,5]. However, first-line pembrolizumab or atezolizumab seemed to be effective for NSCLC patients with a PS of 2 if PD-L1 expression was greater than 50%.

Currently, the administration of a PD-1 blockade in NSCLC patients with a poor PS remains controversial. These problems are discussed in this review.

## 2. PS as a Prognostic Factor after a PD-1 Blockade

Recently, several studies demonstrated that the Eastern Cooperative Oncology Group (ECOG) PS was an independent factor for predicting worse outcomes after PD-1 blockade monotherapy in patients with advanced or recurrent NSCLC [4,5,6,7]. Fujimoto et al. retrospectively analyzed the prognostic significance in 613 patients that were treated with nivolumab in a second-line or over setting [4]. Of the 613 patients, an ECOG PS of 0 or 1 was observed in 472 patients, an ECOG PS of 2 in 94 patients, and an ECOG PS of 3 or 4 in 47 patients. The objective response rate (ORR) of patients with an ECOG PS of 0 or 1, ECOG PS of 2, and ECOG PS of 3 or 4 were 24, 11, and 4%, respectively. A statistically significant difference in the progression-free survival (PFS) was observed between the patients with an ECOG PS of 0 or 1 and an ECOG PS of 2 (*p* < 0.001), and between those with an ECOG PS of 2 and an ECOG PS of 3 or 4 (*p* = 0.022). Their multivariate analysis identified never-smokers, poor ECOG PS, and epidermal growth factor receptor (*EGFR*) mutations or anaplastic lymphoma kinase (*ALK*) rearrangements as independent predictors of worse PFS. Katsura et al. also investigated the difference in efficacy between an ECOG PS of 0–1 (good PS, *n* = 43) and an ECOG PS of 2–4 (poor PS, *n* = 20) in previously treated NSCLC patients receiving nivolumab [6]. The median overall survival (OS) was 412 days for an ECOG PS of 0 or 1, 32 days for an ECOG PS of 2–4, and 31 days in best supportive care (ECOG PS of 0 or 1 vs. ECOG PS of 2–4, *p* < 0.001; ECOG PS of 2–4 vs. best supportive care, *p* = 0.137). Moreover, a statistically significant difference in the ORR was recognized between an ECOG PS of 0 or 1 and an ECOG PS of 2–4 (23% vs. 0%, *p* < 0.001). Imai et al. evaluated the efficacy of first-line pembrolizumab monotherapy in elderly patients with NSCLC with PD-L1 ≥ 50% [3]. Thirty-seven (78.7%) of 47 patients had an ECOG PS of 0 or 1, 7 (15.0%) had an ECOG PS of 2, and 3 (6.3%) had an ECOG PS of 3 or 4. An ECOG PS of 2 or 3 was identified as an independent factor for predicting poor outcomes using multivariate analysis. Other studies also demonstrated that an ECOG PS of 2–4 was identified as an independent factor for predicting worse outcomes in NSCLC patients that were treated with nivolumab or pembrolizumab [5,7]. In previous retrospective studies, the ECOG PS was the most important prognostic factor for predicting worse outcomes after PD-1 blockade treatment in patients with NSCLC. However, the relationship between a worse ECOG PS and the failed efficacy of PD-1 blockade remains unclear. Table 1 shows several studies that examined the prognostic factors in NSCLC patients who received PD-1 blockade.

## 3. Efficacy of a PD-1 Blockade in NSCLC Patients with a PS of 2

Recently, Kano et al. reported the clinical features of a PD-1 blockade in patients with NSCLC with a poor PS [8]. They retrospectively analyzed and compared the prognostic significance after PD-1 blockade initiation in 448 patients with an ECOG PS of 0–1 to 79 patients with an ECOG PS of 2–4. The median PFS was 4.1 months for an ECOG PS of 0–1 and 2.0 months for an ECOG PS of 2–4, with a significant difference (*p* < 0.001). The patients with an ECOG PS of 0–1 exhibited a better OS (median, 17.4 months) than those with an ECOG PS of 2–4 (median, 4.0 months) (*p* < 0.001). In the analysis according to the ECOG PS level, the median PFS was 6.9 months for a PS of 1, 3.5 months for an ECOG PS of 2, 2.3 months for an ECOG PS of 2, and 1.1 months for an ECOG PS of 3–4. Their multivariate analysis also demonstrated that the ECOG PS was an independent predictor of worse outcomes. It is noteworthy that there was no statistically significant difference in the PFS (8.1 months vs. 7.3 months; *p* = 0.321) and OS (reached vs. not reached; *p* = 0.148) between the patients with an ECOG PS of 0–1 and an ECOG PS of 2 harboring PD-L1 ≥ 50%. However, the median PFS (3.5 months vs. 2.0 months; *p* < 0.001) and OS (16.7 months vs. 4.7 months; *p* < 0.001) in patients with any PD-L1 expression displayed a significant difference between patients with an ECOG PS of 0–1 and those with an ECOG PS of 2. The results of this study suggest that a PD-1 blockade is effective in a limited population of NSCLC patients with PD-L1 ≥ 50%. Next, we focus on the efficacy of a PD-1 blockade in NSCLC patients with an ECOG PS of 2 in a previous literature review (Table 2).

In three studies that focused on PD-L1 ≥ 50% in a first-line setting, the median PFS was 4.0 to 12.3 months and the median OS was greater than 7.4 months [8,12,13]. Aside from the high PD-L1 expression, the median PFS yielded 1.2 to 8.3 months, indicating the difference in efficacy of the PD-1 blockade according to the expression of PD-L1 in patients with an ECOG PS of 2 (Table 2). Regarding tumor shrinkage, the ORR of patients with PD-L1 ≥ 50% seemed to be favorable compared with those with any PD-L1 expression. PePS2 was a prospective phase 2 study that investigated the efficacy and safety of pembrolizumab according to PD-L1 expression in NSCLC patients with a PS of 2 [12]. Sixty patients were eligible for the analysis, and the ORR was 21% in first-line patients (*n* = 24) and 31% in subsequent-line patients (*n* = 36); the ORR was 11% in patients with PD-L1 < 1% (*n* = 27), 33% in those with PD-L1 of 1–49% (*n* = 15), and 47% in those with PD-L1 ≥ 50% (*n* = 15). The median PFS and OS were 3.7 and 8.1 months, respectively, in patients with PD-L1 less than 1%, 8.3 and 12.6 months, respectively, in those with PD-L1 of 1–49%, and 12.6 and 14.6 months, respectively, in those with PD-L1 of 50% or greater. Adverse events were recognized in 28% of patients without early death or grade 5 treatment-associated toxicity. The authors concluded that pembrolizumab can be safely administered with comparable efficacy to patients with an ECOG PS of 0–1 with no increase in the occurrence of immune-related toxicities [12]. Patients with an ECOG PS of 2 represent 20–30% of the proportion that is diagnosed with advanced NSCLC and are sometimes candidates for carboplatin doublets, although their prognosis is dismal [14]. The therapeutic efficacy of PD-1 blockade monotherapy in second or further lines for NSCLC patients with an ECOG PS ≥ 2, an ORR of 3 to 11%, a median PFS of less than 2 months, and a median OS of 3.5–6.0 months is disappointing [15]. Considering the results of previous reports, pembrolizumab in a first-line setting serves as a suitable regimen for advanced NSCLC patients with an ECOG PS of 2 if their PD-L1 expression is ≥50%.

## 4. Retrospective Analysis of First-Line Pembrolizumab in PD-L1 ≥ 50% NSCLC with a Poor PS

Pembrolizumab is the standard of care for patients with NSCLC with PD-L1 ≥ 50%. A poor PS remains a strong prognostic factor for ICIs, as described in prospective and retrospective studies. Recently, Facchinetti et al. reported a systematic review and meta-analysis of the clinical usefulness of first-line immunotherapy in NSCLC patients with a poor PS [16]. In their systematic review, 41 studies were chosen, and 1030 of 5357 (19%) of the patients in 30 studies on first-line pembrolizumab presented with an ECOG PS of 2–4 at pembrolizumab initiation. Their meta-analysis demonstrated that the ORR and disease control rate (DCR) in NSCLC patients with an ECOG PS of 2–4 were 30.9 and 41.5%, respectively, compared with 55.2 and 71.5% in those patients with an ECOG PS of 0–1. Moreover, the PFS and OS (at 6, 12, 18, and 24 months) were approximately double in patients with an ECOG PS of 0–1 compared with patients with an ECOG PS of 2–4. It is noteworthy that a poor PS was correlated with a lower incidence of immune-related adverse events (21.2% compared with 35% for an ECOG PS of 0–1), with the possible explanation being the shorter exposure to pembrolizumab in patients with an ECOG PS of 2–4. In real-world data, they found that 19% of patients treated with first-line pembrolizumab had an ECOG PS of 2–4. Furthermore, Tomasik et al. analyzed 26,442 patients in 67 studies. The results of their review demonstrated that in patients with an ECOG PS of 2–4 vs. those with an ECOG PS of 0–1, the pooled odds ratios for the ORR and DCR were 0.46 (95% CI: 0.39–0.54) and 0.39 (95% CI: 0.33–0.48), respectively; furthermore, the pooled hazard ratios for the PFS and OS were 2.17 (95% CI: 1.96–2.39) and 2.76 (95% CI: 0.84–1.48), respectively [17]. This systematic review and meta-analysis indicated a significant reduction in both the ORR and DCR in patients with an ECOG PS of 2–4 compared with those with an ECOG PS of 0–1.

## 5. Interpretation of a Poor PS in NSCLC Patients with Comorbidity

Comorbidities may affect the decreased physical condition in patients with a poor PS. If careful observation is not paid to such patients, we may confuse a worse PS due to cancer progression with that which is secondary to comorbidity. Recently, Facchinetti et al. reported that NSCLC patients with an ECOG PS of 2 due to comorbidities exhibited a significantly better outcome compared with patients with a similar PS that was secondary to tumor progression when first-line pembrolizumab was administered for PD-L1 expression ≥50% [15]. Patients with PD-L1 ≥ 50% having a poor PS that is caused by comorbidities can benefit from first-line pembrolizumab. It was speculated that comorbidities do not reduce the immune response of the host, whereas tumor deterioration harboring intrinsic aggressiveness induced protein catabolism with potential cancer cachexia [15]. Recently, Zeng et al. explored the impact of comorbidity in NSCLC patients undergoing PD-1 blockade, as assessed using the Charlson comorbidity index (CCI) [18]. Their study suggested that the comorbidity burden might be a predictor for prognosis in NSCLC that is treated with immunotherapy [18]. Physicians should be aware of the relationship between comorbidities and poor PS before the initiation of ICI treatment.

## 6. Metabolic Relationship between a Poor PS and a PD-1 Blockade

The ECOG PS is well known to be an established factor for predicting worse outcomes after any treatment in human neoplasms. However, physical activity, such as the ECOG PS, is closely correlated with metabolism determined by glucose, amino acids, and fatty acids, and tumor glucose and amino acid metabolism are reportedly associated with tumor progression, metastasis, and survival [19,20]. Positron emission tomography (PET) imaging with 2-[fluorine-18]-fluoro-2-deoxy-D-glucose (^18^F-FDG) as an alternative assessment of tumor glucose metabolism was established as a significant molecular imaging modality for the diagnosis of lung cancer. Similar to the ECOG PS, the increased accumulation of ^18^F-FDG within cancer cells was described to be closely related to prognostic factors in patients with NSCLC by several reports [20,21]. Recently, several investigations depicted the correlation between PD-L1 expression and ^18^F-FDG uptake within tumor cells in NSCLC [22] and focused on the relationship between the expression of PD-L1 and tumor metabolism. Although ^18^F-FDG PET has the potential to predict the therapeutic efficacy of immunotherapy as a metabolic response, the detailed mechanism of the tumor metabolic response and immunotherapy remains unclear [22,23]. In such a situation, there were several descriptions regarding the relationship between muscle content or the extent of obesity and the efficacy of immunotherapy, and the presence of sarcopenia was negatively correlated with the efficacy of ICIs in NSCLC [24,25]. Several studies demonstrated that a high body mass index (BMI) is closely related to favorable survival after ICI initiation in patients with melanoma, NSCLC, and renal cell carcinoma [24,25,26]. Ichihara et al. analyzed 513 NSCLC patients using a BMI cut-off value of 22 kg/m^2^ and reported that no significant difference in the PFS and OS was observed between the high- and low-BMI patients with NSCLC (*n* = 84) harboring high PD-L1 expression (≥50%) and receiving first-line pembrolizumab; contrarily, the PFS and OS of high-BMI patients who received PD-1 blockade monotherapy as second- or later-line treatment was significantly longer than those of low BMI patients (PFS: 3.7 vs. 2.8 months, *p* = 0.036; OS: 15.4 vs. 13.5 months, *p* = 0.021) (*n* = 429) [27]. Thus, BMI may be associated with the efficacy of ICIs in patients with NSCLC. Although it remains unclear why obesity improves the efficacy of PD-1 blockades in cancer patients, an in vivo study demonstrated that the frequency of PD-1 on tumor-infiltrating CD8^+^ T cells was significantly higher in obese mice than in control mice, and a PD-1 blockade was apparently effective in obese mice compared with control mice [28]. These investigations suggest that the tumor infiltration of PD-1 + CD8^+^ T cells, which is easily responsive to PD-1 blockade, was higher in obese cancer patients compared with those who were not obese. Generally, a poor PS signifies a better prognosis for cancer patients with sarcopenia. We hypothesize that PD-1 + CD8^+^ T cells are decreased within tumor specimens in NSCLC patients with a poor PS, who may be resistant to PD-1 blockades.

## 7. Discussion

Currently, there is no established biomarker for predicting the efficacy and prognosis after PD-1 blockade initiation in patients with NSCLC, in addition to PD-L1 expression within tumor cells. In previous prospective and retrospective studies, the PS was identified as an independent prognostic factor for predicting worse outcomes. Although the NSCLC patients with an ECOG PS of 0–1 depicted a better prognosis after a PD-1 blockade than those with an ECOG PS of 2–4, those with a PS of 2 exhibited a comparable prognosis to those with an ECOG PS of 0–1 in first-line pembrolizumab treatment for the population with a PD-L1 ≥ 50%. Notably, a poor PS resulting from comorbidities was described to have a better prognosis after PD-1 blockade treatment than that due to tumor progression itself. Although the presence of comorbidities may confuse the assessment of ECOG PS status, careful examination is necessary to distinguish comorbidity from cancer progression for clinicians.

The relationship between a poor PS and the lower efficacy of ICIs remains unclear. However, recent evidence demonstrated that metabolic environments, such as obesity, may increase the number of tumor-infiltrating lymphocytes and the efficacy of PD-1 blockade treatment in NSCLC and melanoma patients. Considering the association between obesity and the efficacy of ICIs, the status of a poor PS may be substantially opposite to that of obesity; thus, it is speculated that the tumor immune environment in patients with a decreased PS tends to be resistant to a PD-1 blockade.

As pointed out in previous reports [8,27], ICIs are not effective for NSCLC patients with an ECOG PS of 3–4. Kano et al. described that 15 NSCLC patients with an ECOG PS of 3–4 displayed a median PFS and OS of 1.1 and 1.9 months, respectively [8]. Even if first-line pembrolizumab is administered to patients with a PD-L1 expression ≥50%, the median PFS and OS in patients with an ECOG PS of 3–4 were 1.0 and 2.9 months, respectively, similar to the survival data of best supportive care. Inaba-Higashiyama et al. reported that three patients with an ECOG PS of 3 progressed rapidly under treatment with pembrolizumab [29]. The results of this evidence suggest that first-line pembrolizumab is a non-effective treatment for NSCLC patients with an ECOG PS of 3–4 compared with best supportive care.

Moreover, Santini et al. described that a higher proportion of NSCLC patients who had received ICIs within 30 days of death had a PS ≥ 2 [30]. Jiménez Galán et al. also examined the clinical efficacy of the NSCLC patients receiving pembrolizumab as first-line treatment according to the level of the ECOG PS using a single-center retrospective study [31]. In their study, the ORR, median PFS, and median OS in the patients with an ECOG PS 0–1 and an ECOG PS ≥ 2 were 26.1 and 5.7% (*p* = 0.014), respectively; 9.6 and 1.6 months (*p* < 0.001), respectively; and 18.9 and 2.0 months (*p* < 0.001), respectively [31]. They concluded that the ECOG PS was a single independent predictor of the OS and PFS found in their study. Other researchers agreed with these results showing that ICIs were not effective for NSCLC patients with an ECOG PS ≥ 2 [32,33]. As one mechanism of low response to a PD-1 blockade against the NSCLC patients with a poor PS, the presence of serum vascular endothelial growth factor (VEGF) was suggested by one retrospective study [34]. The high levels of serum VEGF were significantly related to a shorter PFS in patients with an ECOG PS of 2 and the ORR tended to be lower in patients with higher VEGF [34]. Several researchers reported a worse prognostic value for pretreatment levels of some clinical-pathological parameters, such as the neutrophil-to-lymphocyte ratio (NLR) [35], lung immune prognostic index (LIPI) [36], and some composite scores [37,38]. High pretreatment levels of the NLR led to an inferior outcome with nivolumab in patients with pretreated NSCLC [35]. It was reported that a pretreatment LIPI combining NLR and lactate dehydrogenase (LDH) was associated with a worse outcome from using ICIs but not chemotherapy [36]. Park et al. previously described a multivariate risk prediction model, namely, the iSEND, which categorizes advanced or metastatic NSCLC patients treated with PD-1 blockade into good, intermediate, or poor groups, and the OS in patients with the iSEND good was superior to the iSEND poor [37]. Furthermore, Prelaj et al. previously provided the EPSILoN (Eastern Cooperative Oncology Group performance status (ECOG PS), smoking, liver metastases, LDH, NLR) score, which is a clinical/biochemical prognostic score [38]. They reported that EPSILoN may be useful for identifying advanced NSCLC patients who are able to get some clinical benefit from ICIs [38].

Some studies described a negative correlation between some concomitant medications (steroids or antibiotics) and immunotherapy efficacy [39,40,41]. Ricciuti et al. described how NSCLC patients receiving ≥10 mg of prednisone at the time of the ICI initiation exhibited a worse prognosis than those treated with 0 to <10 mg of prednisone [39]. This phenomenon was also supported by a different study [40]. Krief et al. examined whether the early use of antibiotics could affect the efficacy of ICIs in NSCLC patients and found that the early use of antibiotics was associated with a shorter OS with nivolumab [41]. Although the detailed relationship between a poor PS and the efficacy of ICIs remains unclear, one or some of the potential explanations above may be factors causing a poor PS.

The histology of NSCLC is not homogeneous, and adenocarcinoma is different from squamous cell carcinoma. Approximately half of the patients with pulmonary adenocarcinoma have other strong driver mutations that are sensitive to molecule-targeting drugs, which are not present in squamous cell carcinoma. Although we could not find detailed information about the relationship between a poor PS and ICI efficacy stratified by histology, immune reactions may be different based on the histological types. Further investigation should be focused on the clinical efficacy of ICIs in NSCLC patients with a poor PS according to different histologies.

## 8. Conclusions

The clinical benefit of ICIs is still limited to patients with NSCLC who have a favorable PS. However, first-line pembrolizumab seems to be relatively effective in patients with a PS of 2 harboring PD-L1 expression ≥50%. Clinicians should be alert to the presence of comorbidities, which could be mistaken for a poor PS. Further investigation is warranted to elucidate the detailed mechanism between a poor PS and the lower efficacy of ICIs in human neoplasms.

## Figures and Tables

**Table 1 medicina-57-01273-t001:** Independent prognostic factors in NSCLC patients that were treated with a PD-1 blockade.

First Author [Ref.]	No. of Pts	Drug Type (Treatment Line)	PD-L1 (%)	Smoking Yes/No (Patient’s Number)	ECOG PS 0–1/≥2 (Patient’s Number)	Independent Prognostic Factors for Predicting Negative Outcome (Multivariate Analysis)
Imai H. [3]	47	Pembro (1st-line)	≥50%	43/4	37/10	PS (0–1/2–3), smoking (yes/no), response (non-PD/PD)
Fujimoto D. [4]	613	Nivo (2nd-line~)	Any	482/131	472/141	PS (0–1/2–4), smoking (yes/no), driver mutations (yes/no)
Ichiki Y. [7]	44	Nivo or Pembro (1st- or 2nd-line~)	Any	8/36	32/12	PS (0–1/2–4), histology (Ad/Sq), PET (SUV) (SD), WBC (SD), Neutro (SD), NLR (SD), LDH (SD), Alb (SD)
Ahn B. C. [5]	155	Nivo or Pembro (1st- or 2nd-line~)	Any	104/51	121/34	PS (0–1/2–3), PD-L1 (<50%/≥50%), driver mutations (yes/no), liver metastasis(yes/no)
Kano H. [8]	527	Nivo or Pembro (1st- or 2nd-line~)	Any	445/94	448/79	Staging, smoking (yes/no), PS (0–1/2–4), treatment line (1st/2nd)

Abbreviations: Ref., reference; NSCLC, non-small cell lung cancer; Nivo, nivolumab; Pembro, pembrolizumab; PD-L1, programmed death ligand-1; no. of pts, number of patients; PS, performance status; PD, progressive disease; driver mutations, EGFR mutations or ALK translocation; PET, positron emission tomography; SUV, standardized uptake value; SD, standard deviation; WBC, white blood cell; Neutro, neutrophil; NLR, neutrophil-to-lymphocyte ratio; LDH, lactate dehydrogenase; Alb, albumin; Ad, adenocarcinoma; Sq, squamous cell carcinoma.

**Table 2 medicina-57-01273-t002:** Review of ICI blockade efficacy in patients with a PS of 2.

First Author [Ref.]	Study Design	Drug Type	Histology	Treatment Setting	PD-L1 Status	No. of Pts (PS = 2)	ORR (%)	mPFS (Months)	mOS (Months)
Felip E. [9]	Phase 2	Nivo	SQC	2nd-line~	NA	103	2	NA	5.2
Spigel D. R. [10]	Phase 3	Nivo	All-comers	2nd-line~	NA	128	20	NA	4.0
Barlesi F. [11]	Phase 3	Nivo/Ipi	All-comers	1st-line	NA	139	20	3.6	NA
Middleton G. [12]	Phase 2	Pembro	All-comers	1st or 2nd	Yes	60	27	4.4	9.8
					<1%	27	11	3.7	8.1
					1–49%	15	33	8.3	12.6
					≥50%	15	47	12.6	14.6
Fujimoto D. [4]	Retro	Nivo	All-comers	2nd-line~	NA	94	11	1.2	NA
Alessi J. V. [13]	Retro	Pembro	All-comers	1st-line	≥50%	39	25.6	4.0	7.4
Kano H. [8]	Retro	Pembro	All-comers	1st-line	≥50%	11	NA	7.3	NR
		Nivo/Pembro		2nd-line~	NA	53	NA	2.0	4.7

Abbreviations: Nivo, nivolumab; Pembro, pembrolizumab; SQC, squamous cell lung cancer; NA, not applicable; NR, not reached; PD-L1, programmed death-1; no. of pts, number of patients; PS, performance status; ORR, objective response rate; mPFS, median progression-free survival; mOS, median overall survival; ref., reference.

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
