# Peer review of "Clinical Effectiveness of Immune Checkpoint Inhibitors in Non-Small-Cell Lung Cancer with a Poor Performance Status"

_medicina, 2021, doi:10.3390/medicina57111273_

Round 1

Reviewer 1 Report

The authors have positively addressed the issues raised during the first revision round. I think the manuscript is now stronger.

I have no further comments or suggestions 

Reviewer 2 Report

Based on the revised document and clarification of PS and the CCI and histological subtypes the manuscript looks ok. 

This manuscript is a resubmission of an earlier submission. The following is a list of the peer review reports and author responses from that submission.

Round 1

Reviewer 1 Report

The present manuscript provides an overview of the role of ICIs in lung cancer patients with poor performance status.

Some comments:

  • Table 1 and 2 should include the name of the study (or the name of the first author)
  • Some relevant studies have not been discussed and should be necessary included:
    • Alessi JV, et al. J Immunother Cancer 2020
    • Santini D, et al. J Transl Med. 2021
    • Jiménez Galán R, et al. Biology (Basel). 2021
    • Matsubara T, et al. Onco Targets Ther. 2021
    • Gounant V, et al. Cancers (Basel). 2021
    • Shibaki R, et al. Cancer Immunol Immunother. 2020
  • Some studies have reported a poor prognostic value for pretreatment levels of some clinic-pathological parameters, such as NLR (Russo A, et al. Adv Ther 2020), LIPI score (Mezquita L, et al. JAMA Oncol 2018), and some other composite scores (Park W, et al. Br J Cancer 2019; Prelaj A, et al. Cancers 2019). A comment on these studies should be included.
  • Some studies have reported a negative correlation between some concomitant medications (steroid, antibiotics) and ICIs efficacy (Ricciuti B, et al. J Clin Oncol 2019; Arbour KC, et al. J Clin Oncol 2018; Ouaknine Krief J, et al. J Immunother Cancer. 2019 ; Rossi G, et al. Crit Rev Oncol Hematol 2019). One of the potential explanations is the poor performance status of these patients. A comment on this issue would be useful.

Author Response

The present manuscript provides an overview of the role of ICIs in lung cancer patients with poor performance status.

Some comments:

Table 1 and 2 should include the name of the study (or the name of the first author)

Re) Thank you for your generous comments. According to reviewer’s comments, the name of the first author was included in Table 1 and Table 2.

Some relevant studies have not been discussed and should be necessary included:

Alessi JV, et al. J Immunother Cancer 2020

Santini D, et al. J Transl Med. 2021

Jiménez Galán R, et al. Biology (Basel). 2021

Matsubara T, et al. Onco Targets Ther. 2021

Gounant V, et al. Cancers (Basel). 2021

Shibaki R, et al. Cancer Immunol Immunother. 2020

Re) According to reviewer’s generous suggestions, we added your given references in our revised paper, and some discussion was done.

Some studies have reported a poor prognostic value for pretreatment levels of some clinic-pathological parameters, such as NLR (Russo A, et al. Adv Ther 2020), LIPI score (Mezquita L, et al. JAMA Oncol 2018), and some other composite scores (Park W, et al. Br J Cancer 2019; Prelaj A, et al. Cancers 2019). A comment on these studies should be included.

Re) According to reviewer’s generous suggestions, we added your given references in our revised paper, and some discussion was done.

Some studies have reported a negative correlation between some concomitant medications (steroid, antibiotics) and ICIs efficacy (Ricciuti B, et al. J Clin Oncol 2019; Arbour KC, et al. J Clin Oncol 2018; Ouaknine Krief J, et al. J Immunother Cancer. 2019 ; Rossi G, et al. Crit Rev Oncol Hematol 2019). One of the potential explanations is the poor performance status of these patients. A comment on this issue would be useful.

Re) According to reviewer’s generous suggestions, we added your given references in our revised paper, and some discussion was done.

Reviewer 2 Report

The review is very instructive, however confusing. The authors describe PS 0-1 vs 2-3, however do not give a good description of the definition of performance status. Also in the discussion they allude to the PS not being related to comorbidities rather to the tumor progression itself.

In addition NSCLC is not a homogenous disease, adenocarcinoma is very different than squamous cell carcinoma. The patients with adenocarcinoma have other strong driver mutations which are not present in SCC. 

Would have liked to see this stratified by histology, mention about Charlston Comorbidity Index.

Author Response

The review is very instructive, however confusing. The authors describe PS 0-1 vs 2-3, however do not give a good description of the definition of performance status. Also in the discussion they allude to the PS not being related to comorbidities rather to the tumor progression itself.

In addition NSCLC is not a homogenous disease, adenocarcinoma is very different than squamous cell carcinoma. The patients with adenocarcinoma have other strong driver mutations which are not present in SCC. 

Would have liked to see this stratified by histology, mention about Charlston Comorbidity Index.

Re) Thank you for your generous comments.

The definition of PS in our paper is described based on the criteria of ECOG PS. Therefore, we corrected the detailed description of PS in whole manuscript. According to reviewer’s suggestions, some discussions were added in our paper.